# Usefulness of Ultrasound Shear Wave Elastography for Detection of Quadriceps Contracture in Immobilized Rats

**DOI:** 10.3390/ani14010076

**Published:** 2023-12-24

**Authors:** Kanokwan Suwankanit, Miki Shimizu, Kazuhiko Suzuki, Masahiro Kaneda

**Affiliations:** 1Department of Veterinary Diagnostic Imaging, Faculty of Agriculture, Tokyo University of Agriculture and Technology, 3-5-8 Saiwai-cho, Tokyo 183-0054, Japan; s212485w@st.go.tuat.ac.jp; 2Department of Clinical Sciences and Public Health, Faculty of Veterinary Science, Mahidol University, Nakhon Pathom 73170, Thailand; 3Laboratory of Veterinary Toxicology, Faculty of Agriculture, Tokyo University of Agriculture and Technology, 3-5-8 Saiwai-cho, Tokyo 183-0054, Japan; kzsuzuki@cc.tuat.ac.jp; 4Laboratory of Veterinary Anatomy, Faculty of Agriculture, Tokyo University of Agriculture and Technology, 3-5-8 Saiwai-cho, Tokyo 183-8509, Japan; kanedam@cc.tuat.ac.jp

**Keywords:** muscle atrophy, muscle elasticity, muscle stiffness, range of joint motion, sarcomere length

## Abstract

**Simple Summary:**

Quadriceps contracture is a rare disorder that occurs particularly in young animals. Prolonged immobilization of the stifle joint in the extended position is one factor in the development of quadriceps contracture. Clinical diagnostic methods include history taking and physical examination. In most cases, a diagnosis of quadriceps contracture is obtained at an advanced stage of the disease when the prognosis is poor. Therefore, a method for the early clinical detection of quadriceps contracture is needed. We evaluated muscle elastic modulus using ultrasound shear wave elastography, muscle stiffness using a tissue hardness meter, and serum creatinine phosphokinase levels over time in quadriceps contracture model rats and compared their ability to detect pathology. Quadriceps contracture was induced by immobilization of the joints of the hindlimb. During the 4 week immobilization period, in addition to the three clinical examinations, we measured the range of joint motion, histopathology of the muscles, and levels of fibrosis-associated mRNA expression. The elastic modulus increased with the progression of quadriceps contracture, suggesting that ultrasound shear wave elastography is a useful method for the early detection of this disease.

**Abstract:**

Quadriceps contracture is an abnormal pathological shortening of the muscle–tendon unit. To improve the prognosis of quadriceps contracture, improvement of its diagnostic method is needed. In this study, we evaluated the diagnostic utility of ultrasound shear wave elastography in a rat model of quadriceps contracture induced by immobilization. Fifty Wistar rats were randomly divided into control and immobilization groups. During up to 4 weeks of joint immobilization, the quadriceps elastic modulus, muscle hardness, creatinine phosphokinase levels, joint range of motion, histopathologic parameters, and levels of fibrosis-associated mRNA expression were measured every week in the immobilization and control groups and compared. In the immobilization group, the elastic modulus gradually but significantly increased (*p* < 0.05) throughout the immobilization period. However, muscle hardness and serum creatinine phosphokinase levels only increased at 1 and 2 weeks after the start of immobilization, respectively. Muscle atrophy and shortening progressed throughout the immobilization group. Collagen type I and III, α-SMA protein, and mRNA expression of *IL-1β* and *TGF-β1* significantly increased (*p* < 0.05) throughout in the immobilization group. Ultrasound shear wave elastography is the most useful method for clinical assessment of muscle contracture.

## 1. Introduction

Quadriceps contracture is characterized by the replacement of the quadriceps muscle–tendon unit with fibrous tissue, resulting in a shortening of the quadriceps muscle, limiting the range of stifle joint motion, and leading to muscle atrophy and fibrosis [1,2]. The important factor of quadriceps contracture is long-term stifle joint immobilization following orthopedic surgery [3,4]. The prognosis for limb function depends on the degree of quadriceps contracture [3,4,5]: the earlier the diagnosis, the better the prognosis [2]. 

The traditional method of diagnosis of quadriceps contracture is history taking and physical examination [1]. Animals with quadriceps contracture present with chronic lameness and a decreased range of stifle joint motion. Blood biochemistry panels tend to be within the normal range, although creatinine phosphokinase levels may be increased [2]. Atrophy of the quadriceps muscle can cause it to feel like a thickened cord when palpated [1,4]. As the disease progresses, the elasticity of the quadriceps muscle decreases and muscle stiffness increases. Therefore, it is desirable to be able to clinically quantify muscle elasticity and thus detect lesions in the quadriceps muscle. 

Muscle stiffness, which is defined as the change in muscle resistance to a change in muscle length [6], can be assessed using palpation, a tissue hardness meter [7,8], ultrasound elastography [9,10,11], or magnetic resonance elastography [12,13]. Palpation is commonly used to evaluate the resistance of muscle; however, it is a subjective assessment. A tissue hardness meter measures the strength of tissue resistance by applying pressure but cannot evaluate individual muscles. Magnetic resonance elastography can provide a larger measurement field than ultrasound and is not limited by muscle depth [14]. It can also quantify muscle stiffness and provide three-dimensional data [13,15]. However, the difficulty of obtaining magnetic resonance equipment, its high cost, and limited use in monitoring quadriceps contracture are major disadvantages [16].

Ultrasound shear wave elastography is a low-invasive, dynamic, and real-time ultrasound imaging technology [17]. Dependent on the shear wave velocity, the Young’s modulus elasticity value is calculated, which relies on the use of an ultra-high-speed algorithm [18]. This technique can capture the distribution of stiffness within specific muscles along the direction of the muscle by measuring the elastic modulus (in kilopascals) [18,19,20]. It produces both a qualitative color-coded shear wave velocity map placing on the B-mode image inside a region of interest [21] and a quantitative evaluation of tissue elasticity [17,22]. The disadvantage is that there are limits to the depth and range of measurements that can be taken [17]. It has been used to investigate the elastic modulus of relaxed and contracted muscles in humans [19]. However, there have been no reports on the use of shear wave elastography to evaluate quadriceps contracture. In this study, three clinical examinations (ultrasound shear wave elastography, tissue hardness meter, and serum creatinine phosphokinase levels) were performed during a period of immobilization of up to 4 weeks in a rat model of quadriceps contracture. The pathophysiology of quadriceps contracture was assessed by the range of joint motion, histopathology, and gene expression levels of *interleukin-1β* (*IL-1β*), *transforming growth factor-β1* (*TGF-β1*) and *hypoxia-inducible transcription factor-1α* (*HIF-1α*). We hypothesized that ultrasound shear wave elastography would be the best method for detecting quadriceps contracture because of its ability to detect changes in muscle elasticity. 

## 2. Materials and Methods

### 2.1. Animals

Fifty 13 week old, male, specific pathogen-free (SPF) Wistar rats (287.1 ± 12.8) were obtained from Japan SLC International Cooperation, Tokyo, Japan. The animals were maintained in individual cages with dimensions measuring 25 × 40 × 20 cm, with a 12 h light and 12 h dark cycle at a room temperature of 24 ± 2 °C. Food and water were available ad libitum. The experimental protocol was approved by the ethics review committee for animal experimentation at the Tokyo University of Agriculture and Technology (approval number R04-191). Rats were randomly divided into a control group (Group C, *n* = 24) and an immobilization group (Group I, *n* = 26). In each group, the samples were evaluated by equally sampling every other week during the 4 week experimental period (in Group I, weeks 1 and 2, *n* = 6; weeks 3 and 4, *n* = 7).

### 2.2. Immobilization Procedure

All rats were anesthetized with isoflurane, which was administered using the open drop technique. They were maintained in a correct stage of anesthesia with an anesthetic machine using 1–2% isoflurane with an oxygen flow rate of 300–600 mL/min. For the duration of the anesthesia, the muscles were fully relaxed and did not respond to stimuli; each rat’s respiratory rate was maintained at 50–100 bpm. Then the fur on both hindlimbs was clipped. Rats in Group I were immobilized with the stifle joint in full extension and the ankle joint in plantar flexion using a spiral wire immobilization method [23,24]. This method immobilizes the bilateral hindlimbs and is less stressful for the rats than other immobilization methods. To prevent dermatitis, the rats were taped with non-elastic bandage tape (Multipore^TM^ Sports White Athletic Tape, 3M Japan or Kinesio^®^ tape, Tokyo, Japan) to both hindlimbs. A steel bonsai wire (diameter 2.5 mm, length 70 cm) was then applied at the level of the fourth to the fifth lumbar vertebrae and coiled from the hip joints to the end of both hindlimbs. No analgesics or anti-inflammatory drugs were administered during the creation of this model, because no pain or inflammation occurred. The hip joint could move, whereas the stifle and ankle joints could not move. The stifle and ankle joints were immobilized for 1, 2, 3, and 4 weeks. The immobilization procedure was not performed on rats in Group C. The rats were monitored once a day for any sign of bandage loosening or any adverse incidences. The bandages were changed when loosened or an adverse incidence such as hindlimb edema, impaired blood flow at hindlimb, skin injury, or hindlimb necrosis occurred. All of the rats were fitted with vest collars; these were modified versions of the vest collars developed by Jang et al. (2016) [25]. 

### 2.3. Joint Angle and Range of Motion Measurement

Flexion and extension angles of the hip, stifle, and ankle joints were measured under anesthesia with isoflurane before and every week during the 4 week experimental period. Joint angles were evaluated using a goniometer according to the method of Millis and Levine (2013) [26]. The range of joint motion was calculated by deducting the flexion angle from the extension angle of that joint.

### 2.4. Measuring of the Quadriceps Muscle Thickness and Elastic Modulus

Quadriceps thickness and elastic modulus were measured before testing and every week up to 4 weeks during the experimental period. Ultrasonography (LOGIQ-E9 2.0 ultrasound system, GE health care^®^, Wauwatosa, WI, USA) was performed under anesthesia with isoflurane. The rats were placed in a supine position and the stifle joint angle was set at 135° using a template. Measurements were taken at the quadriceps muscle belly in the right and left limbs. The transducer was set in the vertical direction to the surface of the muscle. Sufficient transmission gel was used to avoid pressure on the muscle by the transducer, to reduce air between the rat skin and the transducer, and to obtain clear images. Echogenicity of muscle fibers is low; perimysium and fascia are highly echogenic.

The thickness of the quadriceps muscle was evaluated by B-mode ultrasound scanning using an L8-18i linear probe (bandwidth: 8–18 MHz; footprint: 35.0 × 7.0 mm). Muscle thickness was defined as the length of a line drawn on a transverse image at 50% of the length of the femur, from the muscle surface to the femur, through the center of the rectus femoris muscle [27,28]. Three measurements were taken, and the mean value was calculated. 

Muscle elastic modulus was measured in the rectus femoris, vastus lateralis, and vastus medialis muscle by ultrasound shear wave elastography. The probe (9L linear transducer; bandwidth: 2.5–8 MHz; footprint: 49.0 × 9.0 mm) was applied to the muscle belly, and muscle fibers were delineated parallel to the probe’s long axis direction. A square region of interest (1 × 1 cm) was placed within the muscle, and a color-coded elasticity map was displayed in the box in real-time. The scaling of the color display ranged from red to blue, with red indicating high elastic modulus or stiffest tissue and blue indicating low elastic modulus or softest tissue. The fascia and vessel are not included in the region of interest. A measurement circle range (5 mm in diameter for the rectus femoris and 2 mm in diameter for the vastus lateralis and the vastus medialis muscles) was set at the center of the square region of interest. Five measurements were taken, and the mean value was calculated. 

### 2.5. Measuring Muscle Hardness

Quadriceps hardness was measured using a PEK-MP tissue hardness meter (Imoto Machinery, Tokyo, Japan) before testing and every week during the 4 week experimental period. Stiffness (Newtons, N) is measured on a scale from 0 (soft) to 100 (hard). The rats were placed in a supine position and the stifle joint was extended to 135° [29]. Three measurements were taken at the midpoint of quadriceps of both limbs, and the average value was used for the measurements. 

### 2.6. Evaluation of Creatinine Phosphokinase Level 

Creatinine phosphokinase levels were measured weekly during the 4 week experimental period. Blood was collected by cardiac puncture while the rats were alive [30]. Blood was placed in heparin tubes and centrifuged to obtain serum. Creatinine phosphokinase levels were measured on a DRI-CHEM NX700 analyzer (Fujifilm Medical Co., Ltd., Tokyo, Japan).

### 2.7. Muscle Collection and Measurement

Every week during the 4 week experimental period, rats were euthanized using the open drop technique with 30% isoflurane. Bilateral thoracotomy was performed to confirm cardiac arrest according to the Guide for the Care and Use of Laboratory Animals [31,32]. Then, bilateral quadriceps muscles were excised, wet weight was measured using digital scales and quadriceps muscle weight was standardized by body weight. Muscle size (length × width × height) was measured using a digital caliper.

### 2.8. Histopathological Examination

The belly of the right quadriceps muscle was cut into a longitudinal section and a cross-section. Both sections were fixed in 4% paraformaldehyde at 4 °C for 36–48 h. These specimens were washed in distilled water and dehydrated in a graded series of ethanol. They were then equilibrated with xylene and embedded in paraffin. Serial cross-sectional and longitudinal 4 µm sections were cut using a microtome and then stained with hematoxylin & eosin (H&E) [33]. 

A light microscope (Keyence BZ-X800, Keyence Corporation, Osaka, Japan) was used to take photographs for the histological findings at a final magnification of ×400, and ImageJ software version 1.53 (National Institutes of Health, Bethesda, MD, USA) was used to perform the morphological analysis. Sarcomere length was measured by taking 10 longitudinal images per quadriceps muscle, and more than 1000 sarcomeres were randomly selected. Cross-sectional area, minimum Feret’s diameter, perimeter, and diameter of muscle fibers were measured by scanning cross-sectional images and randomly selecting 400 muscle fibers.

### 2.9. Immunohistochemical Analysis

For the immunohistochemical analysis, longitudinal sections of the right quadriceps muscle were deparaffinized in xylene, rehydrated with a graded series of ethanol, and washed in running water. Then, the sections were autoclaved in a target retrieval solution at 121 °C for 10 min. To inhibit endogenous peroxidase activity, the sections were incubated with 1% H_2_O_2_ in methanol for 30 min at room temperature, before being washed in 0.01 M phosphate-buffered saline (PBS; pH 7.4) and then blocked with 1% normal goat serum for 30 min at room temperature. The sections were incubated overnight at 4 °C with collagen type I monoclonal antibody (COL-1; Invitrogen^®^, Carlsbad, CA, USA; diluted ×200), collagen type III monoclonal antibody (FH-7A, Invitrogen^®^, Rockford, IL, USA; diluted ×50), and monoclonal mouse anti-human smooth muscle actin (Clone 1A4, Sigma-Aldrich^®^, Glostrup, Denmark; diluted ×200). Then, the sections were rinsed in PBS for 15 min, after which they were incubated with EnVision horseradish peroxidase anti-mouse (Dako EnVision+, HRP, Agilent Technology^®^, Santa Clara, CA, USA) for 30 min at room temperature. The sections were rinsed in PBS, and horseradish peroxidase-binding sites were visualized with 0.05% 3,3′-diaminobenzidine and 0.01% H_2_O_2_ in 0.05 M Tris buffer at room temperature. After a final washing step, the sections were counterstained with hematoxylin. Finally, the sections were dehydrated in increasing concentrations of alcohol, washed with xylene, and mounted on slides. Immunohistochemical analysis of the samples was performed using a light microscope (Keyence BZ-X800, Keyence Corporation, Osaka, Japan) in combination with ImageJ software version 1.53 (National Institutes of Health). The number of myofibroblasts was determined from the images by measuring the α-smooth muscle actin (α-SMA)-positive cells. The number of α-SMA-positive cells was counted in all fields for each section at ×200 magnification, and the average number per field was calculated. The levels of collagen types I and III were determined using a semi-quantitative technique. The percentage of collagen to total tissue longitudinal area, including the muscle, perivascular, and interstitial areas, was calculated for the total slide area at ×200 magnification. Areas of muscle overlap or poor staining were excluded. All analyses were performed under conditions where the analyst was blinded to the experimental group to which the slides belonged.

### 2.10. Real-Time RT-PCR to Measure Interleukin-1β (IL-1β), Transforming Growth Factor-β1 (TGF-β1), and Hypoxia-Inducible Transcription Factor-1α (HIF-1α) mRNA

The left quadriceps muscle was used for real-time RT-PCR analysis. Total RNA was extracted from the muscle samples using TRIzol reagent (TRIzol™, Invitrogen^®^, Carlsbad, CA, USA), according to the manufacturer’s protocol. Total RNA was used as a template with the qPCR RT Master Mix (ReverTra Ace, Toyobo Co., Ltd., Osaka, Japan) to prepare cDNA. The mRNA expression levels were then determined using real-time RT-PCR, which was performed in an optical 96 well plate with the StepOnePlus^TM^ Real-Time PCR System (Applied BioSystems, Foster City, CA, USA). The primer sequences used for the real-time RT-PCR are shown in Table 1. Relative quantification was calculated using the 2^−∆∆Ct^ method [34] and normalized to β-actin. Data are presented as expression levels relative to the expression level in the control group.

### 2.11. Statistical Analysis

GraphPad Prism version 8 (GraphPad Software Inc., San Diego, CA, USA) was used to perform the statistical analysis. All data are presented as mean and standard deviation. Comparisons between control and immobilization groups in the same period were performed using the Mann–Whitney test, while differences in each period within the same group were assessed using the Kruskal–Wallis test, followed by Dunn’s multiple comparison test. Statistical significance was set at *p* < 0.05.

## 3. Results

### 3.1. Joint Angle and Range of Motion (ROM)

The measured angles of maximum extension, maximum flexion, and range of motion of the hip, stifle, and ankle joints are shown in Table 2. There was no significant difference in the range of motion of the hip joint between Group C and Group I for each period. In Group I, the stifle and ankle flexion angles of Group I increased (*p* = 0.0001) and the range of motion of both joints decreased (*p* = 0.0001). The range of motion of the stifle and ankle joints in Group I at 3 and 4 weeks after immobilization was decreased compared to Group I at 1 week.

### 3.2. Measurement of Thigh Muscle Thickness 

Representative measurements of thigh muscle thickness are shown in Figure 1. The hypoechogenicity of muscle fibers and hyperechogenicity of perimysium and fascia were observed. 

The measurements of quadriceps muscle thicknesses are shown in Table 3. The quadriceps muscle thickness in Group I was decreased in all periods compared to Group C. In Group C, on the other hand, the quadriceps muscle thickness increased as the rats grew.

### 3.3. Elastic Modulus

Representative longitudinal ultrasound shear wave elastography images of the rectus femoris, vastus lateralis, and vastus medialis muscle are shown in Figure 2, Figure 3, and Figure 4, respectively. In Group I, cyan, orange, and red colors increased during the immobilization period, suggesting an increased elastic modulus.

The elastic modulus of the rectus femoris, vastus lateralis, and vastus medialis muscle as determined by ultrasound shear wave elastography are shown in Figure 5A–C. The elastic modulus of the rectus femoris, vastus lateralis, and vastus medialis muscles in Group C were 8.6 ± 1.0, 7.4 ± 1.3, 7.6 ± 1.1 kPa (mean and standard deviation) in before experiment, 7.5 ± 1.4, 7.8 ± 0.8, 8.7 ± 0.5 kPa in 1 week, 7.4 ± 1.2, 7.1 ± 1.6, 8.0 ± 1.5 kPa in 2 weeks, 8.6 ± 1.2, 7.5 ± 0.9, 8.4 ± 0.7 kPa in 3 weeks, and 8.0 ± 1.0, 7.4 ± 1.0, 8.5 ± 0.7 kPa in 4 weeks, respectively. The elastic modulus of those muscles in Group I was 8.3 ± 1.2, 7.4 ± 1.1, 7.7 ± 1.1 kPa in the before experiment, 15.0 ± 1.4, 15.2 ± 2.2, 14.5 ± 1.5 kPa in 1 week, 20.1 ± 1.7, 19.6 ± 2.2, 18.9 ± 2.5 kPa in 2 weeks, 29.0 ± 2.1, 28.8 ± 1.7, 29.2 ± 2.4 kPa in 3 weeks, and 30.6 ± 2.7, 32.9 ± 4.0, 33.1 ± 3.4 kPa in 4 weeks, respectively. The elastic modulus for all periods in Group I was significantly higher (*p* = 0.0001 in all muscles) compared with the same periods in Group C. In Group I, the elastic modulus increased after more than 3 weeks of immobilization compared to the 1 and 2 week immobilizations. 

### 3.4. Muscle Hardness

The quadriceps muscle hardness values are shown in Figure 5D. The quadriceps muscle hardness of Group C and Group I were 67 ± 4.3 and 65 ± 3.2 N (mean and standard deviation) before the experiment, 65 ± 5.1 and 69 ± 4.2 N in 1 week, 66 ± 5.3 and 66 ± 3.3 N in 2 weeks, 66 ± 4.7 and 68 ± 5.6 N in 3 weeks, and 67 ± 6.5 and 68 ± 4.9 N in 4 weeks. The hardness value of the quadriceps muscles was only higher in Group I compared with Group C in week 1 (*p* = 0.0424).

### 3.5. Creatinine Phosphokinase Level

The creatinine phosphokinase levels are shown in Table 4. In Group I, the level of creatinine phosphokinase increased at 2 weeks compared with Group C in the same period (*p* = 0.0152).

### 3.6. The Ratio of Muscle Weight to Body Weight and Quadriceps Muscle Measurement

The ratio of muscle weight to body weight and the length, width, and height of quadriceps muscles are shown in Table 5. These measurements were lower in Group I compared with Group C in the same time periods. 

### 3.7. Histological Evaluation of Quadriceps Muscle Fiber and Sarcomere Length

Histological findings of the quadriceps muscle cross-sections stained with H&E for all groups showed a normal microanatomic arrangement, with no lesions of note in the muscle fibers and no inflammatory cells between the muscle bundles. In Group I, the muscle bundles were separated as muscle fascicles surrounded by disorganized and thickened connective tissue in the perimysium and endomysium. 

The data for the cross-sectional area, minimum Feret’s diameter, diameter, perimeter, and sarcomere length of the quadriceps muscles is shown in Table 6. The cross-sectional area, minimum Feret’s diameter, diameter, and perimeter of Group I decreased (*p* = 0.0001) compared with these values for Group C in the same period and showed a gradual decrease (*p* = 0.0001) with increasing immobilization time. The sarcomere length in Group I was lower (*p* = 0.0001) compared with Group C in the same period and decreased as the immobilization period increased. 

### 3.8. Measurement of α-Smooth Muscle Actin (α-SMA) Cells and Collagen Types I and III

The number of α-SMA-positive cells and the percentage of collagen types I and III are shown in Table 7. These measurements in Group I were significantly higher than those in Group C in the same period and increased as the immobilization period increased. 

### 3.9. Relative Expression of IL-1β, TGF-β1, and HIF-1α mRNA

The expression levels of *IL-1β*, *TGF-β1*, and *HIF-1α* mRNA in the quadriceps muscle are shown in Figure 6. *IL-1β* mRNA expression in Group I in the 1, 2, 3, and 4 week periods was 3.4 ± 0.8 fold, 2.7 ± 1.6 fold, 2.4 ± 0.8 fold, and 2.7 ± 1.3 fold (mean and standard deviation) higher, respectively, compared with the levels in Group C in the same periods (Figure 6A). *TGF-β1* mRNA expression in the 1, 2, 3, and 4 week periods was 1.6 ± 0.3 fold, 1.9 ± 0.8 fold, 2.3 ± 1.1 fold, and 2.4 ± 1.5 fold higher, respectively, compared with the levels in Group C in the same periods (Figure 6B). *HIF-1α* mRNA expression in Group I in the 1, 2, 3, and 4 week periods was 1.0 ± 0.2 fold, 1.0 ± 0.2 fold, 1.1 ± 0.3 fold, and 1.8 ± 0.5 fold, respectively, compared with the levels in Group C in the same periods (Figure 6C). In Group I, the expression levels of *IL-1β* and *TGF-β1* mRNA were significantly higher than those in Group C in each of the time periods. The expression level of *HIF-1α* mRNA in Group I at 4 weeks after immobilization was significantly higher than the other measurement periods in Group I and Group C in the same period.

## 4. Discussion

We evaluated the effectiveness of ultrasound shear wave elastography and the measurement of muscle hardness and creatinine phosphokinase level to diagnose quadriceps muscle contracture by comparing each of them with the pathogenesis of quadriceps contracture. After immobilization for 1, 2, 3, and 4 weeks, the elastic modulus of three muscles in the quadriceps group, the rectus femoris, vastus lateralis, and vastus medialis muscles, increased as quadriceps contracture progressed. However, the hardness value of the quadriceps muscles only increased in the period 1 week after immobilization. The serum creatinine phosphokinase level only increased in the period 2 weeks after immobilization. Therefore, we found that ultrasound shear wave elastography was a more accurate method to detect the pathophysiology of quadriceps contracture than measuring muscle hardness or serum creatinine phosphokinase levels.

Quadriceps contracture is defined as quadriceps atrophy, with muscle tissue replaced by fibrous tissue, resulting in muscle shortening [1,2]. In this study, for all periods of immobilization, decreased muscle size and length were observed. Our histological analysis showed that the atrophic changes included reductions in the cross-sectional area, minimum Feret’s diameter, diameter, and perimeter of quadriceps muscle fiber as it progressed through 4 weeks of immobilization. Sarcomere length also decreased in all periods in the immobilization group. Our histological and morphological analyses showed that muscle fiber atrophy and muscle shortening result from quadriceps muscle contracture. These results were consistent with the previously reported characteristics of muscle contracture induced by muscle immobilization in the shortened position [35,36]. 

The mechanism of muscle fibrosis in muscle contracture is related to several factors that increase collagen production. These factors include *IL-1β* and *TGF-β1*. Previous research has shown that *IL-1β* induces *TGF-β1* synthesis via a TGF-β-dependent mechanism [37]. This induces fibroblast activation and collagen production [38]. *TGF-β1* plays an important role in activating the differentiation of fibroblasts into myofibroblasts, a process related to muscle fibrosis [38,39]. In our study, the expression of *IL-1β* and *TGF-β1* mRNA in the immobilization group was higher than that in the control group. This result is similar to the trend reported from a previous study [36]. Another factor associated with fibrosis is hypoxia. The expression of *HIF-1α* increases in fibrotic tissue and is related to hypoxia [40,41]. Hypoxia is stimulated by a decrease in the number of capillaries; this decrease is induced after immobilization for 4 weeks [42]. Honda et al. (2015) [36] showed that *HIF-1α* mRNA expression increased 4 weeks after immobilization and promoted muscle fibrosis in cases of muscle contracture. In our research, we found that in the immobilization group, *HIF-1α* mRNA expression was increased at 4 weeks after immobilization compared with 1, 2, or 3 weeks after immobilization. Previous research has also shown that α-SMA is related to the fibrosis mechanism [43]. α-SMA is a marker of myofibroblasts [44]. Myofibroblasts produce large quantities of collagen and are an important factor in pathological muscle contracture [45,46]. We found that the number of α-SMA positive cells in the immobilization group was increased at 1 week after immobilization. The overexpression of collagen types I and III in muscles, including in the perimysium, endomysium, and epithelium, is the main characteristic of muscle fibrosis in muscle contracture induced by immobilization [23,47]. In our study, the percentages of collagen types I and III also increased from 1 week after immobilization. We assume that muscle fibrosis in quadriceps contracture occurred during the early stages of immobilization (less than 2 weeks after immobilization), before progressing in the later stages of immobilization (3–4 weeks after immobilization). These results are similar to previously reported results [36]. Together, these results indicate that the upregulation of *IL-1β*, *TGF-β1*, and *HIF-1α* mRNA affects the conversion of fibroblasts into myofibroblasts and induces collagen production, which is associated with the occurrence of quadriceps muscle contracture. 

Gait assessment and palpation of quadriceps contracture are conventional assessment methods. Myogenic contractures occur after less than 2 weeks of joint immobilization, and muscle contractures with arthrogenic and myogenic changes occur after more than 4 weeks of joint immobilization [48]. Lameness due to quadriceps contracture occurs 3 to 5 weeks after injury [1]. When arthrogenic contracture occurs, the prognosis for treatment to return to normal function is poor [4]. 

A decreased range of motion in the stifle joint and a taut band around the cranial thigh muscle are the clinical signs of quadriceps contracture [1]. This can occur if the stifle joint is immobilized in the extension position for a long time [1,2]. In our study, we found that the range of stifle and ankle joint motion in the immobilization group decreased compared with that in the control group. These results are similar to those described in previous studies [1,3,4,23]. We assume that quadriceps muscle contracture was induced in all periods of the immobilization. The decreased stifle joint range of motion associated with progressive quadriceps contracture may be useful in detecting this condition. However, muscle properties cannot be objectively assessed.

Serum creatinine phosphokinase is another biomarker that can be used to diagnose muscle damage and myopathy [2,49]. Creatinine phosphokinase is an enzyme present in skeletal muscle tissue. When muscle is damaged, the enzyme leaks into the bloodstream [50]. However, a previous study of sartorius muscle contracture noted that there were no serum biochemical tests that are specific for muscle contracture [51]. In our research, we found that the level of serum creatinine phosphokinase only showed an increase at 2 weeks after immobilization, which means that it cannot be used to diagnose quadriceps contracture. The quadriceps contracture in this study was not due to muscle injury, but to abnormal shortening of the muscle–tendon unit induced by immobilization. Thus, it was suggested that serum creatinine phosphokinase levels do not increase after immobilization. 

In recent years, it has been shown that imaging technologies can play a major role in the diagnosis of myopathies [17,52,53,54,55]. B-mode ultrasonography can be used to evaluate muscle atrophy and decreased muscle size [28]. In the present study, the quadriceps muscle thickness of the immobilized group was decreased. These results were consistent with the findings of muscle atrophy on histopathological examination. B-mode ultrasonography is a less invasive technique than histopathologic evaluation and can be used to assess muscle atrophy in the clinical practice. 

Ultrasound shear wave elastography can evaluate muscle trauma and pathological changes in musculoskeletal tissue [17,55]. Although there have been reports of ultrasound elastography being used to assess muscle hardness and stiffness [56], there is a lack of information about the use of the technique to diagnose quadriceps contracture. Ultrasound shear wave elastography is a method that can be used to measure muscle stiffness and assess the changes in muscle elasticity that are characteristic of muscle contracture. In our study, we found there was a trend for quadriceps contracture lesions to have a high elasticity modulus and increased muscle stiffness when compared with pre-experimental data and the control group. The elastic modulus and muscle stiffness increased gradually with the progression of muscle contracture. Guo et al. (2018) [57] used ultrasound shear wave elastography to study three patients with gluteal muscle contracture and concluded that the elastic modulus of gluteal muscle with contracture is higher than that of the normal muscle. The increased elastic modulus, which indicates decreased muscle elasticity, was consistent with overexpression of collagen types I and III and increased expression levels of *IL-1β* and *TGF-β1* mRNA, suggesting muscle fibrosis associated with progressive muscle contracture. Therefore, ultrasound shear wave elastography is a diagnostic method that can assess muscle properties compared to joint range of motion and is less invasive than histopathology or gene expression level examination.

Tissue hardness meters can be used to detect muscle stiffness; however, their use in diagnosing quadriceps contracture is controversial. Niitsu et al. (2011) [8] reported that after exercise, muscles exhibit increased muscle hardness, which they measured using a tissue hardness meter. However, Akagi et al. (2015) [11] investigated neck and shoulder stiffness using both ultrasound elastography and a tissue hardness meter, but they found that the tissue hardness meter could not evaluate the muscle hardness level as well as ultrasound elastography. In our study, we found that the muscle hardness level increased only after 1 week in the immobilization group compared with that of the control group. Therefore, a tissue hardness meter cannot be used to detect quadriceps contracture. Decreased muscle elasticity is related to changes in the fibril arrangement in the endomysium as muscle contracture progresses [35]. Okita et al. (2004) [35] reported that collagen fibrils in the endomysium during muscle contracture were oriented in the circumferential direction at short sarcomere length. Consequently, based on quadriceps contracture pathogenesis involving the shortening of muscle length and decrease in muscle elasticity resulting in muscle stiffness, ultrasound shear wave elastography is more effective in quantifying muscle elasticity and muscle stiffness than a tissue hardness meter, which can only assess muscle stiffness. 

Our study had several limitations. First, we did not evaluate the elastic modulus of the vastus intermedius muscle because this muscle is very small, and an ultrasound probe cannot access it. Second, the ultrasound elastography results were not compared with other imaging techniques, such as magnetic resonance imaging; this was because in this study we focused on diagnostic methods that can be easily accessed in clinical practice. Third, we only used male rats, in accordance with previous studies of muscle contracture [23]. It is unclear what effect sex hormones in the estrus cycle might have on the induction of muscle contracture, although it is known that estrogen and progesterone in the estrus cycle can affect muscle maintenance and increase muscle mass [58,59]. 

## 5. Conclusions

A diagnosis of quadriceps contracture can be obtained by palpating the muscle and assessing the gait status. In such cases, the disease has already advanced, and there is little chance of recovering muscle function. Furthermore, the level of serum creatinine phosphokinase and a tissue hardness meter cannot assess the pathogenesis of quadriceps contracture. The range of stifle joint motion can be used to evaluate quadriceps contracture, but it cannot provide objective assessments of muscle properties. Histopathology and expression levels of *IL-1β* and *TGF-β1* mRNA can assess the pathology and properties of the muscle but are more invasive techniques than ultrasound shear wave elastography. Shear wave elastography can be used to detect changes in muscle elasticity in quadriceps contracture after just 1 week of immobilization. Therefore, shear wave elastography is a potentially useful method for evaluating quadriceps contracture in the early stage. When the disease is diagnosed earlier and treatment started as soon as possible, the prognosis of quadriceps contracture is improved. Ultrasound shear wave elastography is expected to contribute to future research and have applications in clinical practice for early treatment and improving the prognosis of muscle contracture. 

## Figures and Tables

**Figure 1 animals-14-00076-f001:**
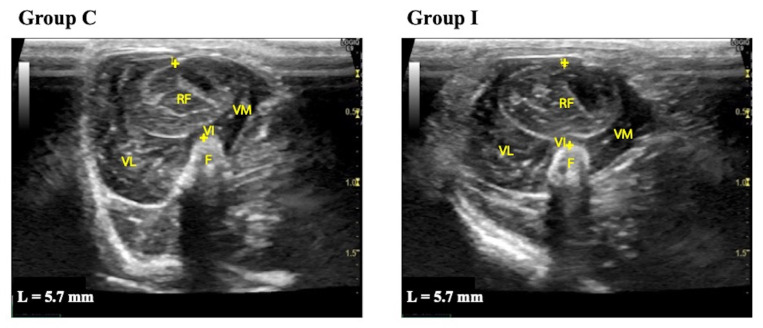
Representative transverse B-mode ultrasound images of the quadriceps muscle used to measure muscle thickness before the experiment. Group C, control group; Group I, immobilization group; L, thigh muscle thickness. F: femur; RF: rectus femoris; VL: vastus lateralis; VM: vastus medialis; VI: vastus intermedius; +: electronic caliper. The images show the placement of the electronic calipers to measure the cross-section at 50% of the femur length by passing across the center of the rectus femoris muscle from the surface of the muscle to the femoral bone.

**Figure 2 animals-14-00076-f002:**
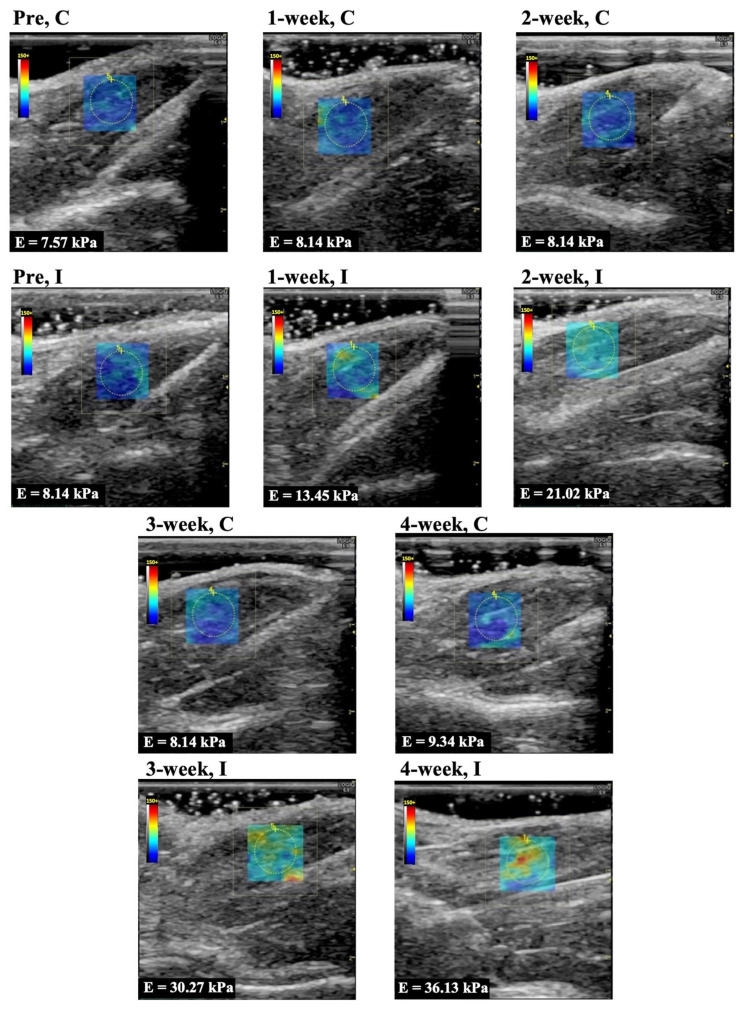
Representative shear wave elastography of the rectus femoris muscle. C, control group; I, immobilization group; Pre, before the experiment; E, elastic modulus. The rectangle indicates the region of interest. The measurement circle was 5 mm in diameter.

**Figure 3 animals-14-00076-f003:**
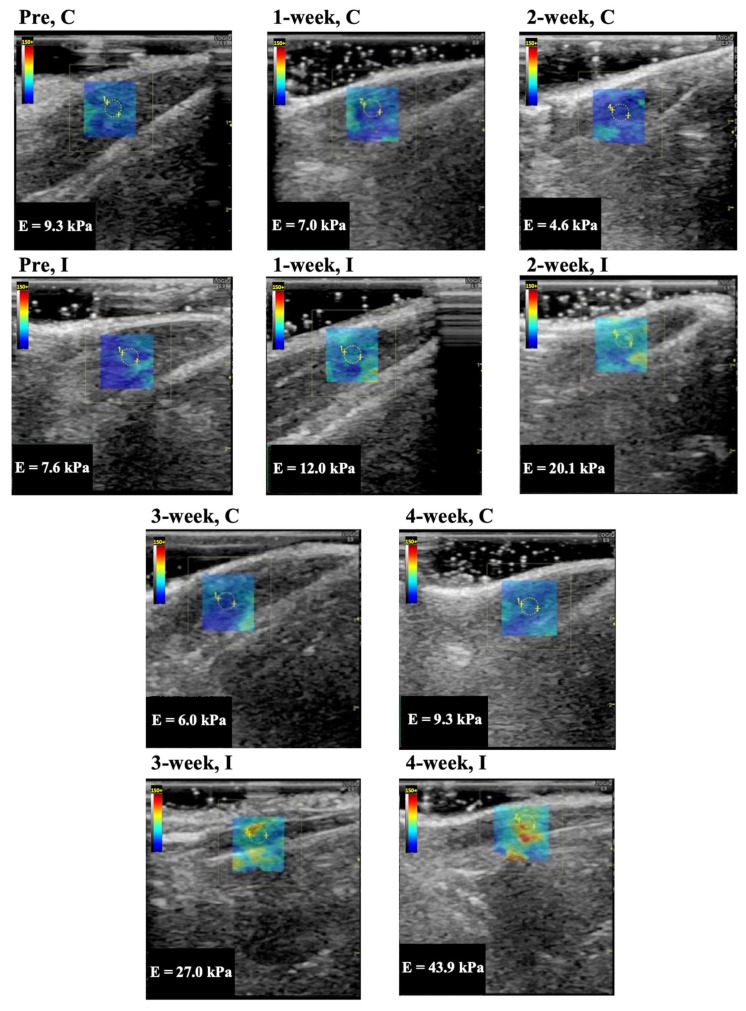
Representative shear wave elastography of the vastus lateralis muscle. C, control group; I, immobilization group; Pre, before the experiment; E, elastic modulus. The rectangle indicates the region of interest. The measurement circle was 2 mm in diameter.

**Figure 4 animals-14-00076-f004:**
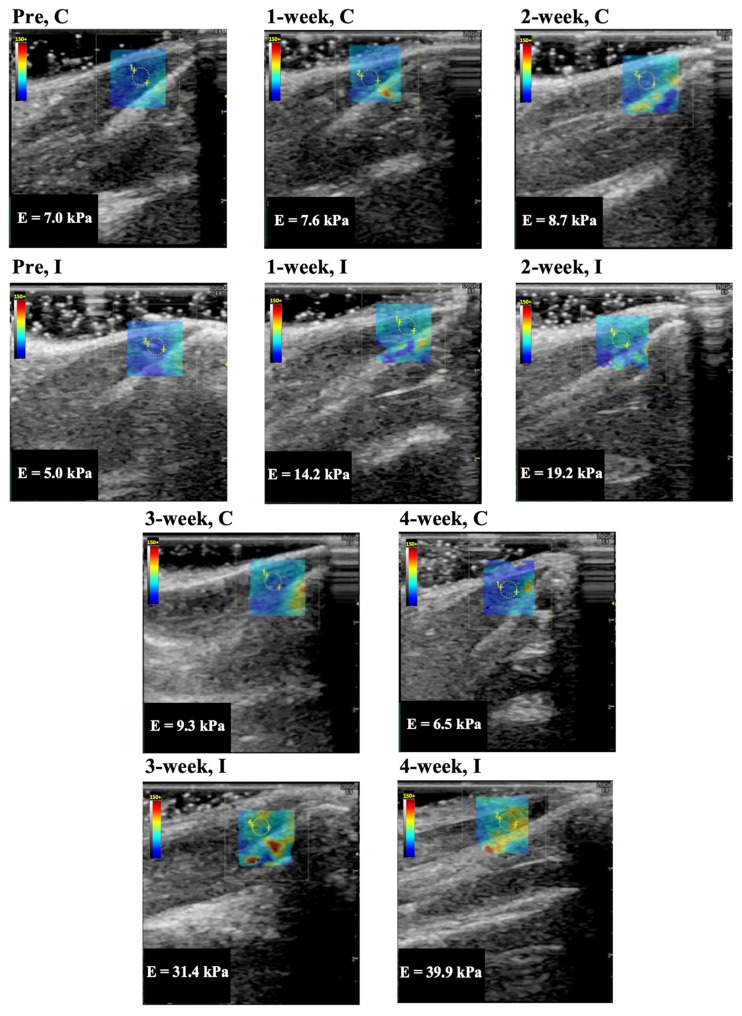
Representative shear wave elastography of the vastus medialis muscle. C, control group; I, immobilization group; pre, before the experiment; E, elastic modulus. The rectangle indicates the region of interest. The measurement circle was 2 mm in diameter.

**Figure 5 animals-14-00076-f005:**
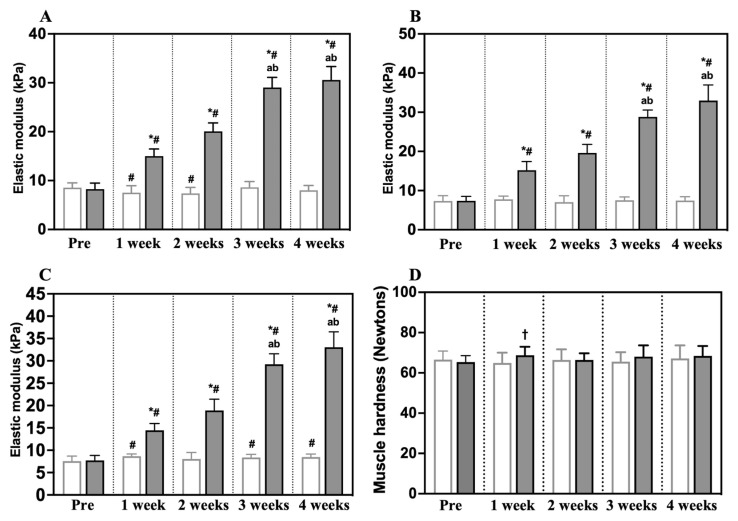
The elastic modulus of the rectus femoris (**A**), vastus lateralis (**B**), and vastus medialis muscles (**C**), measured by ultrasound shear wave elastography; and the quadriceps muscle hardness (**D**), measured using a tissue hardness meter. The empty (white) bars show Group C (control group); the gray bars show Group I (immobilization group); Pre, before the experiment; * significantly different from Group C in the same period (*p* < 0.0001); ^†^ significantly different from Group C in the same period (*p* < 0.05); ^#^ significantly different from Pre in the same group (*p* < 0.05); ^a^ significantly different from the 1 week period in the same group (*p* < 0.001); ^b^ significantly different from the 2 week period in the same group (*p* < 0.05). Values shown are mean and standard deviation.

**Figure 6 animals-14-00076-f006:**
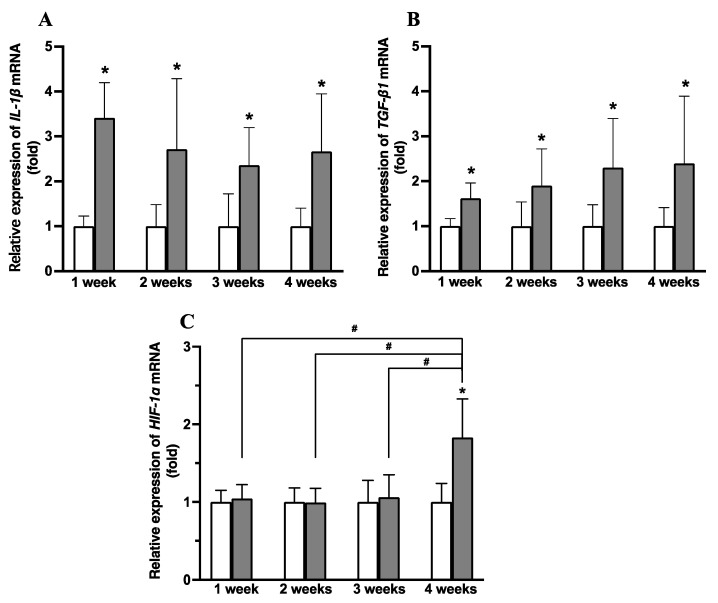
Expression levels of *IL-1β* (**A**), *TGF- β1* (**B**), and *HIF-1α* (**C**) mRNA. The open (white) bars show the control groups, and the gray bars show the immobilization groups. Values are expressed as mean and standard deviation. * Significantly different from Group C in the same period (*p* < 0.05); ^#^ significantly different between periods in the immobilization group (*p* < 0.05).

**Table 1 animals-14-00076-t001:** Sequences of primers used for the real-time RT-PCR.

Object Gene	F/R	Sequence
*IL-1β*	F	5′-AATGACCTGTTCTTTGAGGCTGAC-3′
	R	5′-CGAGATGCTGCTGTGAGATTTGAA-3′
*TGF-β1*	F	5′-ATGCCAACTTCTGTCTGGGG-3′
	R	5′-GGTTGTAGAGGGCAAGGACC-3′
*HIF-1α*	F	5′-TGCTGGCTCCCTATATCCCA-3′
	R	5′-GGAGGGCTTGGAGAATTGCT-3′
*β-actin*	F	5′-GCAGGAGTACGATGAGTCCG-3′
	R	5′-ACGCAGCTCAGTAACAGTCC-3′

F, forward; R, reverse.

**Table 2 animals-14-00076-t002:** Joint angle (°) and ROM of rats in the control and immobilization groups.

Joint and Position		Duration	Group C	Group I
Hip	Extension	Pre	156 ± 3.2 (*n* = 48)	157 ± 3.3 (*n* = 52)
		1 week	155 ± 2.3 (*n* = 12)	156 ± 3.1 (*n* = 12)
		2 weeks	155 ± 1.4 (*n* = 12)	156 ± 1.9 (*n* = 12)
		3 weeks	156 ± 1.7 (*n* = 12)	156 ± 1.4 ^#^ (*n* = 14)
		4 weeks	156 ± 2.2 (*n* = 12)	157 ± 2.5 (*n* = 14)
	Flexion	Pre	30 ± 1.6 (*n* = 48)	31 ± 2.0 (*n* = 52)
		1 week	30 ± 1.0 (*n* = 12)	30 ± 0.6 (*n* = 12)
		2 weeks	30 ± 0.8 (*n* = 12)	31 ± 1.7 (*n* = 12)
		3 weeks	30 ± 0.9 (*n* = 12)	30 ± 1.3 (*n* = 14)
		4 weeks	30 ± 0.6 (*n* = 12)	31 ± 4.0 (*n* = 14)
	ROM	Pre	126 ± 3.3 (*n* = 48)	127 ± 3.4 (*n* = 52)
		1 week	126 ± 2.4 (*n* = 12)	126 ± 3.2 (*n* = 12)
		2 weeks	126 ± 1.5 (*n* = 12)	126 ± 2.5 (*n* = 12)
		3 weeks	127 ± 1.8 (*n* = 12)	125 ± 2.0 (*n* = 14)
		4 weeks	127 ± 2.6 (*n* = 12)	126 ± 3.9 (*n* = 14)
Stifle	Extension	Pre	164 ± 2.4 (*n* = 48)	164 ± 2.5 (*n* = 52)
		1 week	162 ± 2.4 ^#^ (*n* = 12)	162 ± 3.2 ^#^ (*n* = 12)
		2 weeks	164 ± 2.0 (*n* = 12)	162 ± 2.5 ^#^ (*n* = 12)
		3 weeks	162 ± 2.3 ^#^ (*n* = 12)	161 ± 2.1 ^#^ (*n* = 14)
		4 weeks	163 ± 2.4 (*n* = 12)	163 ± 2.5 (*n* = 14)
	Flexion	Pre	30 ± 1.3 (*n* = 48)	30 ± 2.0 (*n* = 52)
		1 week	30 ± 0.8 (*n* = 12)	50 ± 8.4 *^#^ (*n* = 12)
		2 weeks	30 ± 0.3 (*n* = 12)	72 ± 11.8 *^#^ (*n* = 12)
		3 weeks	30 ± 0.0 (*n* = 12)	96 ± 7.5 *^#a^ (*n* = 14)
		4 weeks	30 ± 0.0 (*n* = 12)	112 ± 7.9 *^#ab^ (*n* = 14)
	ROM	Pre	134 ± 2.8 (*n* = 48)	134 ± 3.3 (*n* = 52)
		1 week	132 ± 2.6 (*n* = 12)	112 ± 8.4 *^#^ (*n* = 12)
		2 weeks	134 ± 2.1 (*n* = 12)	89 ± 11.1*^#^ (*n* = 12)
		3 weeks	132 ± 2.3 ^#^ (*n* = 12)	66 ± 6.9 *^#a^ (*n* = 14)
		4 weeks	133 ± 2.4 (*n* = 12)	52 ± 8.1 *^#ab^ (*n* = 14)
Ankle	Extension	Pre	167 ± 2.6 (*n* = 48)	167 ± 3.2 (*n* = 52)
		1 week	169 ± 1.5 ^#^ (*n* = 12)	168 ± 3.3 (*n* = 12)
		2 weeks	168 ± 2.4 (*n* = 12)	170 ± 2.6 ^#^ (*n* = 12)
		3 weeks	168 ± 2.5 (*n* = 12)	169 ± 2.1 ^#^ (*n* = 14)
		4 weeks	170 ± 1.9 ^#^ (*n* = 12)	170 ± 1.8 ^#^ (*n* = 14)
	Flexion	Pre	10 ± 1.3 (*n* = 48)	10 ± 1.4 (*n* = 52)
		1 week	10 ± 0.8 (*n* = 12)	62 ± 14.7 *^#^ (*n* = 12)
		2 weeks	9 ± 1.1 (*n* = 12)	84 ± 13.9 *^#^ (*n* = 12)
		3 weeks	10 ± 0.6 (*n* = 12)	106 ± 6.9 *^#a^ (*n* = 14)
		4 weeks	10 ± 0.4 (*n* = 12)	130 ± 11.2 *^#ab^ (*n* = 14)
	ROM	Pre	158 ± 2.6 (*n* = 48)	158 ± 3.5 (*n* = 52)
		1 week	160 ± 1.3 ^#^ (*n* = 12)	106 ± 13.7 *^#^ (*n* = 12)
		2 weeks	159 ± 2.4 (*n* = 12)	86 ± 13.6 *^#^ (*n* = 12)
		3 weeks	159 ± 2.3 (*n* = 12)	63 ± 7.6 *^#a^ (*n* = 14)
		4 weeks	160 ± 2.1 ^#^ (*n* = 12)	40 ± 11.3 *^#ab^ (*n* = 14)

Values are expressed as mean and standard deviation. Group C, control group; Group I, immobilization group; Pre, before the experiment; * significantly different from Group C in the same period and the same position of the joint (*p* < 0.0001); ^#^ significantly different from Pre in the same group and the same position of the joint (*p* < 0.05); ^a^ significantly different from the 1 week period in the same group and the same position of the joint (*p* < 0.001); ^b^ significantly different from the 2 week period in the same group and the same position of the joint (*p* < 0.05).

**Table 3 animals-14-00076-t003:** Quadriceps muscle thickness measurements (mm).

Duration	Group C	Group I
Pre	5.6 ± 0.3 (*n* = 48)	5.7 ± 0.4 (*n* = 52)
1 week	5.6 ± 0.3 (*n* = 12)	4.8 ± 0.4 *^#^ (*n* = 12)
2 weeks	5.8 ± 0.3 ^#^ (*n* = 12)	4.7 ± 0.4 *^#^ (*n* = 12)
3 weeks	5.9 ± 0.3 ^#^ (*n* = 12)	4.5 ± 0.2 *^#^ (*n* = 12)
4 weeks	6.1 ± 0.2 ^#a^ (*n* = 12)	4.5 ± 0.3 *^#^ (*n* = 14)

Values are expressed as mean and standard deviation. Group C, control group; Group I, immobilization group; Pre, before the experiment; * significantly different from Group C in the same period (*p* < 0.0001); ^#^ significantly different from Pre in the same group (*p* < 0.05); ^a^ significantly different from the 1 week period in the same group (*p* < 0.001).

**Table 4 animals-14-00076-t004:** Creatinine phosphokinase level (U/L).

Duration	Group C	Group I
1 week	345 ± 157 (*n* = 6)	460 ± 383 (*n* = 6)
2 weeks	259 ± 86 (*n* = 6)	475 ± 112 * (*n* = 6)
3 weeks	473 ± 292 (*n* = 6)	493 ± 339 (*n* = 7)
4 weeks	237 ± 99 (*n* = 6)	279 ± 102 (*n* = 7)

Values are expressed as mean and standard deviation. Group C, control group; Group I, immobilization group; * significantly different from Group C in the same period (*p* < 0.05).

**Table 5 animals-14-00076-t005:** Weight and dimensions of quadriceps muscle.

Quadriceps Muscle	1 Week	2 Weeks	3 Weeks	4 Weeks
Muscle weight/body weight (mg/g)				
Group C	8.3 ± 0.4 (*n* = 12)	7.9 ± 0.2 (*n* = 12)	8.4 ± 0.2 ^b^ (*n* = 12)	8.2 ± 0.3 (*n* = 12)
Group I	7.6 ± 0.9 * (*n* = 12)	6.1 ± 0.5 *^a^ (*n* = 12)	6.0 ± 0.3 *^a^ (*n* = 14)	5.9 ± 0.7 *^a^ (*n* = 14)
Muscle length (mm)				
Group C	29.6 ± 1.8 (*n* = 12)	28.9 ± 1.0 (*n* = 12)	29.8 ± 1.4 (*n* = 12)	31.1 ± 2.0 ^b^ (*n* = 12)
Group I	25.7 ± 1.8 * (*n* = 12)	22.9 ± 1.0 *^a^ (*n* = 12)	23.0 ± 1.2 *^a^ (*n* = 14)	22.3 ± 0.8 *^a^ (*n* = 14)
Muscle width (mm)				
Group C	14.3 ± 0.7 (*n* = 12)	14.1 ± 0.9 (*n* = 12)	14.6 ± 1.4 (*n* = 12)	15.2 ± 0.9 (*n* = 12)
Group I	12.4 ± 0.7 * (*n* = 12)	11.7 ± 1.1 * (*n* = 12)	11.5 ± 0.9 * (*n* = 14)	11.3 ± 1.1 *^a^ (*n* = 14)
Muscle height (mm)				
Group C	11.6 ± 1.1 (*n* = 12)	12.0 ± 1.7 (*n* = 12)	12.1 ± 1.1 (*n* = 12)	11.7 ± 1.2 (*n* = 12)
Group I	10.7 ± 1.2 (*n* = 12)	9.0 ± 1.8 *^a^ (*n* = 12)	8.8 ± 0.6 *^a^ (*n* = 14)	8.7 ± 0.5 *^a^ (*n* = 14)

Values are expressed as mean and standard deviation. Group C, control group; Group I, immobilization group; * significantly different from Group C in the same period (*p* < 0.0001); ^a^ significantly different from the 1 week period in the same group (*p* < 0.05); ^b^ significantly different from the 2 week period in the same group (*p* < 0.05).

**Table 6 animals-14-00076-t006:** Cross-sectional area, minimum Feret’s diameter, diameter, perimeter, and sarcomere length of quadriceps muscles analyzed using H&E staining.

Quadriceps Muscle	1 Week	2 Weeks	3 Weeks	4 Weeks
Cross-sectional area (µm^2^)				
Group C	2586 ± 561 (*n* = 2400)	3907 ± 688 ^a^ (*n* = 2400)	4401 ± 624 ^ab^ (*n* = 2400)	4619 ± 748 ^abc^ (*n* = 2400)
Group I	2236 ± 500 *(*n* = 2400)	1237 ± 336 *^a^ (*n* = 2400)	1091 ± 290 *^ab^ (*n* = 2800)	1029 ± 271 *^abc^ (*n* = 2800)
Minimum Feret’s diameter (µm)				
Group C	57.9 ± 6.8 (*n* = 2400)	59.7 ± 7.4 ^a^ (*n* = 2400)	63.4 ± 7.1 ^ab^ (*n* = 2400)	65.7 ± 7.6 ^abc^ (*n* = 2400)
Group I	45.4 ± 6.5 * (*n* = 2400)	33.4 ± 5.7 *^a^ (*n* = 2400)	31.8 ± 5.3 *^ab^ (*n* = 2800)	31.1 ± 4.9 *^abc^ (*n* = 2800)
Diameter (µm)				
Group C	67.4 ± 5.2 (*n* = 2400)	70.3 ± 6.1 ^a^ (*n* = 2400)	74.7 ± 5.3 ^ab^ (*n* = 2400)	76.4 ± 6.2 ^abc^ (*n* = 2400)
Group I	53.0 ± 5.9 * (*n* = 2400)	39.3 ± 5.4 *^a^ (*n* = 2400)	37.0 ± 4.9 *^ab^ (*n* = 2800)	35.9 ± 4.7 *^abc^ (*n* = 2800)
Perimeter (µm)				
Group C	235.8 ± 21.6 (*n* = 2400)	242.2 ± 23.2 ^a^ (*n* = 2400)	254.1 ± 21.7 ^ab^ (*n* = 2400)	258.3 ± 24.5 ^abc^ (*n* = 2400)
Group I	184.3 ± 21.5 * (*n* = 2400)	136.6 ± 19.3 *^a^ (*n* = 2400)	128.7 ± 17.9 *^ab^ (*n* = 2800)	121.9 ± 17.0 *^abc^ (*n* = 2800)
Sarcomere length (nm)				
Group C	1559 ± 144 (*n* = 6000)	1671 ± 154 ^a^ (*n* = 6000)	1673 ± 127 ^a^ (*n* = 6000)	1699 ± 128 ^abc^ (*n* = 6000)
Group I	1415 ± 146 * (*n* = 6000)	1267 ± 121 *^a^ (*n* = 6000)	1137 ± 105 *^ab^ (*n* = 7000)	1123 ± 93 *^ab^ (*n* = 7000)

Values are expressed as mean and standard deviation. Group C, control group; Group I, immobilization group; * significantly different from Group C in the same period (*p* < 0.0001); ^a^ significantly different from 1 week in the same group (*p* < 0.05); ^b^ significantly different from the 2 week period in the same group (*p* < 0.05); ^c^ significantly different from the 3 week period in the same group (*p* < 0.05).

**Table 7 animals-14-00076-t007:** Number of α-smooth muscle actin (α-SMA)-positive cells and the percentage of collagen type I and III in quadriceps muscles of rats in the control and immobilization groups, based on the immunohistochemistry analyses.

Quadriceps Muscle	1 Week	2 Weeks	3 Weeks	4 Weeks
α-SMA (number of cells/field)				
Group C	1.7 ± 0.2 (*n* = 6)	1.5 ± 0.1 (*n* = 6)	1.6 ± 0.3 (*n* = 6)	1.5 ± 0.3 (*n* = 6)
Group I	5.1 ± 0.7 * (*n* = 6)	5.5 ± 0.4 * (*n* = 6)	8.2 ± 0.7 *^a^ (*n* = 7)	8.7 ± 1.2 *^ab^ (*n* = 7)
Collagen type I (%)				
Group C	0.27 ± 0.10 (*n* = 6)	0.20 ± 0.09 (*n* = 6)	0.21 ± 0.10 (*n* = 6)	0.17 ± 0.05 (*n* = 6)
Group I	1.09 ± 0.29 * (*n* = 6)	1.63 ± 0.17 * (*n* = 6)	2.70 ± 0.37 *^a^ (*n* = 7)	3.74 ± 0.30 *^ab^ (*n* = 7)
Collagen type III (%)				
Group C	0.06 ± 0.02 (*n* = 6)	0.06 ± 0.01 (*n* = 6)	0.06 ± 0.01 (*n* = 6)	0.05 ± 0.00 (*n* = 6)
Group I	0.80 ± 0.07 * (*n* = 6)	1.28 ± 0.08 * (*n* = 6)	1.89 ± 0.09 * (*n* = 7)	2.51 ± 0.24 *^ab^ (*n* = 7)

Values are expressed as mean and standard deviation. Group C, control group; Group I, immobilization group; * significantly different from Group C in the same period (*p* < 0.05). ^a^ significantly different from the 1 week period in the same group (*p* < 0.05); ^b^ significantly different from the 2 week period in the same group (*p* < 0.05).

## Data Availability

The data presented in this study are available in article.

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
