# Peer review of "Usefulness of Ultrasound Shear Wave Elastography for Detection of Quadriceps Contracture in Immobilized Rats"

_animals, 2023, doi:10.3390/ani14010076_

Round 1
Reviewer 1 Report
Comments and Suggestions for Authors
The data presented is interesting and deserves attention. It is important to highlight relevant points that address different techniques. The animal model used is acceptable and can be replicated for different species. The authors should adequately address all highlighted and pointed out points, such as comments. An elaborate reformulation should address the results, conclusions, and references. The tables require special attention, evaluating their maintenance or highlighting of their results throughout the text.

The data presented is interesting and deserves attention. It is important to highlight relevant points that address different techniques. The animal model used is acceptable and can be replicated for different species. The authors should adequately address all highlighted and pointed out points, such as comments. An elaborate reformulation should address the results, conclusions, and references. The tables require special attention, evaluating their maintenance or highlighting of their results throughout the text.
Author Response
We greatly appreciate the time and effort given by the reviewers to improve our manuscript. Based on the reviewer’s comments, we have made some changes. Please see the attachment.

Reviewer 2 Report
Comments and Suggestions for Authors
Firstly, congratulations to the authors for such a complete work. I would just like to make some considerations about this interesting paper.
- To begin with, at the Simple summary, in line 15, could you clarify the literature where it is stated that this type of contracture is the most common in the canine species? Since he later shows this statement again, and in my opinion, it is a very severe limiting pathology but it does not seem to me to be the most frequent contracture observed in the clinic.
-In line 98, maybe it might be better to explain this sentence in another way, since it is a bit difficult to understand the selection protocol of rats.
- In line 102, I would change the words "a surgical stage of anesthesia" to "a correct stage of anesthesia".
- During the immobilization procedure, could you explain why you immobilize both hindlimbs? Wouldn't it have been better to immobilize only one limb and not result in such an aggressive procedure that greatly limits the quality of life of these animals?
On the other hand, they do not comment on the use of analgesic or anti-inflammatory drugs during the procedure. Do they use them? - In line 174, change "Evert" for "Every". - In section "Muscle collection and measurement", Is performing a bilateral thoracotomy strictly necessary to confirm cardiac arrest? - In section 2.10, which is the reason you chose the left quadriceps only? - In section 3.3 you make a statement about the colors observed during the immobilization period. I think it would be interesting to include in Materials and Methods a brief sentence explaining the meaning of the colors for a better understanding of the reader. - In Figure 2, I don't know if it is possible to increase the font size within the images, since it is a little difficult to read. In general, I find it an interesting research article, since it is very complete, dealing with many diagnostic aspects of contracture, however, it does not seem to me that its clinical relevance is very great and therefore, its extrapolation to the canine species, as that the diagnosis of contracture of this quadriceps muscle, as well described in the discussion, is very simple to make, with a simple examination of the patient (gait and palpation).Author Response
We greatly appreciate the time and effort given by the reviewers to improve our manuscript. Based on the reviewer’s comments, we have made some changes. Please see the attachment.

Round 2
Reviewer 1 Report
Comments and Suggestions for Authors
The authors did not provide a rationale for not addressing the needs. The paper presents confusing and insufficiently elucidating data. Therefore, I suggest that the authors address the considerations or indeed clarify the points raised.

The authors did not provide a rationale for not addressing the needs. The paper presents confusing and insufficiently elucidating data. Therefore, I suggest that the authors address the considerations or indeed clarify the points raised.
Author Response
Thank you very much for your comments. We greatly appreciate the time and effort given by the reviewers to improve our manuscript. We revised the manuscript based on your comments. Please see the attachment.
